# Functional Equivalence and Path Connectivity of Reducible Hyperbolic Tangent Networks

**Matthew Farrugia-Roberts**
School of Computing and Information Systems
The University of Melbourne
matthew@far.in.net

## Abstract

Understanding the learning process of artificial neural networks requires clarifying the structure of the parameter space within which learning takes place. A neural network parameter's *functional equivalence class* is the set of parameters implementing the same input–output function. For many architectures, almost all parameters have a simple and well-documented functional equivalence class. However, there is also a vanishing minority of *reducible* parameters, with richer functional equivalence classes caused by redundancies among the network's units.

In this paper, we give an algorithmic characterisation of unit redundancies and reducible functional equivalence classes for a single-hidden-layer hyperbolic tangent architecture. We show that such functional equivalence classes are piecewise-linear path-connected sets, and that for parameters with a majority of redundant units, the sets have a diameter of at most 7 linear segments.

## 1   Introduction

Deep learning algorithms construct neural networks through a local search in a high-dimensional parameter space. This search is guided by the shape of some loss landscape, which is in turn determined by the link between neural network parameters and their input–output functions. Thus, understanding the link between parameters and functions is key to understanding deep learning.

It is well known that neural network parameters often fail to uniquely determine an input–output function. For example, exchanging weights between two adjacent hidden units generally preserves functional equivalence (Hecht-Nielsen, 1990). For many architectures, almost all parameters have a simple class of functionally equivalent parameters. These classes have been characterised for multi-layer feed-forward architectures with various nonlinearities (e.g., Sussmann, 1992; Albertini et al., 1993; Kůrková and Kainen, 1994; Phuong and Lampert, 2020; Vlačić and Bölcskei, 2021).

However, all existing work on functional equivalence excludes from consideration certain measure zero sets of parameters, for which the functional equivalence classes may be richer. One such family of parameters is the so-called *reducible parameters*. These parameters display certain structural redundancies, such that the same function could be implemented with fewer hidden units (Sussmann, 1992; Vlačić and Bölcskei, 2021), leading to a richer functional equivalence class.

Despite their atypicality, reducible parameters may play an important role in deep learning. Learning exerts non-random selection pressure, so measure zero sets of parameters may still arise in practice, and indeed reducible parameters are appealing solutions due to parsimony (cf. Farrugia-Roberts, 2023). These parameters are also a source of information singularities (cf. Fukumizu, 1996), relevant to statistical theories of deep learning (Watanabe, 2009; Wei et al., 2022). Moreover, the structure of functional equivalence classes has implications for the local and global shape of the loss landscape, in ways that may influence learning dynamics *near* reducible parameters.

37th Conference on Neural Information Processing Systems (NeurIPS 2023).

In this paper, we study functional equivalence classes for single-hidden-layer networks with the hyperbolic tangent nonlinearity, building on the foundational work of Sussmann (1992) on reducibility in this setting. We offer the following theoretical contributions.

1. In Section 4, we give a formal algorithm producing a canonical representative parameter from any functional equivalence class, by systematically eliminating all sources of structural redundancy. This extends prior algorithms that only handle irreducible parameters.

2. In Section 5, we invert this canonicalisation algorithm to characterise the functional equivalence class of any parameter as a union of simple parameter manifolds. This characterisation extends the well-known result for irreducible parameters.

3. We show that in the reducible case, the functional equivalence class is a piecewise-linear path-connected set—that is, any two functionally equivalent reducible parameters are connected by a piecewise linear path comprising only equivalent parameters (Theorem 6.1).

4. We show that if a parameter has a high degree of reducibility (in particular, if the same function can be implemented using half of the available hidden units), then the number of linear segments required to connect any two equivalent parameters is at most 7 (Theorem 6.3).

While the single-hidden-layer hyperbolic tangent architecture is not immediately relevant to modern deep learning, it enables the first comprehensive analysis of neural network structural redundancy. Moreover, feed-forward layers are a fundamental building block of many modern architectures, and so our analysis is partially informative for such extensions. In Section 7 we discuss such extensions and others, as well as connections to other topics in deep learning including loss landscape analysis, model compression, and singular learning theory.

## 2  Related Work

Sussmann (1992) studied functional equivalence in single-hidden-layer hyperbolic tangent networks, showing that two irreducible parameters are functionally equivalent if and only if they are related by simple operations of exchanging and negating the weights of hidden units. This result was later extended to architectures with a broader class of nonlinearities (Albertini et al., 1993; Kůrková and Kainen, 1994), to architectures with multiple hidden layers (Fefferman and Markel, 1993; Fefferman, 1994), and to certain recurrent architectures (Albertini and Sontag, 1992, 1993a,b,c). More recently, similar results have been found for ReLU networks (Phuong and Lampert, 2020; Bona-Pellissier et al., 2021; Stock and Gribonval, 2022), and Vlačić and Bölcskei (2021, 2022) have generalised Sussmann's results to a very general class of architectures and nonlinearities. However, all of these results have come at the expense of excluding from consideration certain measure zero subsets of parameters with richer functional equivalence classes.

A similar line of work has documented the global symmetries of the parameter space—bulk transformations of the entire parameter space that preserve all implemented functions. The search for such symmetries was launched by Hecht-Nielsen (1990). Chen et al. (1993, also Chen and Hecht-Nielsen, 1991) showed that in the case of multi-layer hyperbolic tangent networks, all analytic symmetries are generated by unit exchanges and negations. Rüger and Ossen (1997) extended this result to additional sigmoidal nonlinearities. The analyticity condition excludes discontinuous symmetries acting selectively on, say, reducible parameters with richer equivalence classes (Chen et al., 1993).

Rüger and Ossen (1997) provide a canonicalisation algorithm. Their algorithm negates each hidden unit's weights until the bias is positive, and then sorts each hidden layer's units into non-descending order by bias weight. This algorithm is invariant precisely to the exchanges and negations mentioned above, but fails to properly canonicalise equivalent parameters that differ in more complex ways.

To our knowledge, there is one line of work bearing directly on the topic of the functional equivalence classes of reducible parameters. Fukumizu and Amari (2000) and Fukumizu et al. (2019) have catalogued methods of adding a single hidden unit to a neural network while preserving the network's function, and Şimşek et al. (2021) have extended this work to consider the addition of multiple hidden units. Though derived under a distinct framing, it turns out that the subsets of parameter space accessible by such unit additions correspond to functional equivalence classes, similar to those we study (though in a slightly different architecture). We note these similarities, especially regarding our contributions (2) and (3), in Remarks 5.4 and 5.5 and Remark 6.2.

## 3 Preliminaries

We consider a family of fully-connected, feed-forward neural network architectures with a single input unit, a single biased output unit, and a single hidden layer of $h \in \mathbb{N}$ biased hidden units with the hyperbolic tangent nonlinearity $\tanh(z) = (e^z - e^{-z})/(e^z + e^{-z})$. Such an architecture has a parameter space $\mathcal{W}_h = \mathbb{R}^{3h+1}$. Our results generalise directly to networks with multi-dimensional inputs and outputs, as detailed in Appendix A.

The weights and biases of the network's units are encoded in the parameter vector in the format $(a_1, b_1, c_1, \ldots, a_h, b_h, c_h, d) \in \mathcal{W}_h$ where for each hidden unit $i = 1, \ldots, h$ there is an *outgoing weight* $a_i \in \mathbb{R}$, an *incoming weight* $b_i \in \mathbb{R}$, and a *bias* $c_i \in \mathbb{R}$, and $d \in \mathbb{R}$ is an *output unit bias*. Thus each parameter $w = (a_1, b_1, c_1, \ldots, a_h, b_h, c_h, d) \in \mathcal{W}_h$ indexes a mathematical function $f_w : \mathbb{R} \to \mathbb{R}$ defined as follows:

$$f_w(x) = d + \sum_{i=1}^{h} a_i \tanh(b_i x + c_i).$$

Two parameters $w \in \mathcal{W}_h, w' \in \mathcal{W}_{h'}$ are *functionally equivalent* if and only if $f_w = f_{w'}$ as functions on $\mathbb{R}$ (that is, $\forall x \in \mathbb{R}, f_w(x) = f_{w'}(x)$). Functional equivalence is of course an equivalence relation on $\mathcal{W}_h$. Given a parameter $w \in \mathcal{W}_h$, the *functional equivalence class* of $w$, denoted $\mathfrak{F}[w]$, is the set of all parameters in $\mathcal{W}_h$ that are functionally equivalent to $w$:

$$\mathfrak{F}[w] = \{ w' \in \mathcal{W}_h \,|\, f_w = f_{w'} \}.$$

For this family of architectures, the functional equivalence class of almost all parameters is a discrete set fully characterised by simple *unit negation and exchange transformations* $\sigma_i, \tau_{i,j} : \mathcal{W}_h \to \mathcal{W}_h$ for $i, j = 1, \ldots, h$, where

$$\sigma_i(a_1, b_1, c_1, \ldots, a_h, b_h, c_h, d) = (a_1, b_1, c_1, \ldots, -a_i, -b_i, -c_i, \ldots, a_h, b_h, c_h, d)$$
$$\tau_{i,j}(a_1, b_1, c_1, \ldots, a_h, b_h, c_h, d) = (a_1, b_1, c_1, \ldots, c_{i-1}, a_j, b_j, c_j, a_{i+1},$$
$$\ldots, c_{j-1}, a_i, b_i, c_i, a_{j+1}, \ldots, a_h, b_h, c_h, d).$$

More formally, these transformations generate the full functional equivalence class for all so-called irreducible parameters (Sussmann, 1992). A parameter $w = (a_1, b_1, c_1, \ldots, a_h, b_h, c_h, d) \in \mathcal{W}_h$ is *reducible* if and only if it satisfies any of the following conditions (otherwise, $w$ is *irreducible*):

   (i)  $a_i = 0$ for some $i$, or
   (ii) $b_i = 0$ for some $i$, or
   (iii) $(b_i, c_i) = (b_j, c_j)$ for some $i \neq j$, or
   (iv) $(b_i, c_i) = (-b_j, -c_j)$ for some $i \neq j$.

Sussmann (1992) also showed that in this family of architectures, reducibility corresponds to *non-minimality*: a parameter $w \in \mathcal{W}_h$ is reducible if and only if $w$ is functionally equivalent to some $w' \in \mathcal{W}_{h'}$ with fewer hidden units $h' < h$. We define the *rank* of $w$, denoted $\mathrm{rank}(w)$, as the minimum number of hidden units required to implement $f_w$:

$$\mathrm{rank}(w) = \min \{ h' \in \mathbb{N} \,|\, \exists w' \in \mathcal{W}_{h'}; \, f_w = f_{w'} \}.$$

Finally, we make use of the following notions of connectivity for a set of parameters. Given a set $W \subseteq \mathcal{W}_h$, define a *piecewise linear path in $W$* as a continuous function $\rho : [0, 1] \to W$ comprising a finite number of linear segments. Two parameters $w, w' \in \mathcal{W}_h$ are *piecewise-linear path-connected in $W$*, denoted $w \leftrightsquigarrow w'$ (with $W$ implicit), if there exists a piecewise linear path in $W$ such that $\rho(0) = w$ and $\rho(1) = w'$. Note that $\leftrightsquigarrow$ is an equivalence relation on $W$. A set $W \subseteq \mathcal{W}_h$ is itself *piecewise-linear path-connected* if and only if $\leftrightsquigarrow$ is the full relation, that is, all pairs of parameters in $W$ are piecewise linear path-connected in $W$.

The *length* of a piecewise linear path is the number of maximal linear segments comprising the path. The *distance* between two piecewise linear path-connected parameters is the length of the shortest path connecting them. The *diameter* of a piecewise linear path-connected set is the largest distance between any two parameters in the set.

# 4 Parameter Canonicalisation

A parameter *canonicalisation algorithm* maps each parameter in a functional equivalence class to a consistent representative parameter within that class. A canonicalisation algorithm serves as a theoretical test of functional equivalence. In Section 5 we invert our canonicalisation algorithm to characterise the functional equivalence class. More practically, canonicalising parameters before measuring the distance between them yields a metric that is independent of functionally irrelevant details such as unit permutations.

Prior work has described canonicalisation algorithms for certain irreducible parameters (Rüger and Ossen, 1997); but when applied to functionally equivalent reducible parameters, such algorithms may fail to produce the same output. We introduce a canonicalisation algorithm that properly canonicalises both reducible and irreducible parameters, based on similar negation and sorting stages, combined with a novel *reduction* stage. This stage effectively removes or 'zeroes out' redundant units through various operations that exploit the reducibility conditions. This process isolates a functionally equivalent but irreducible subparameter.

**Algorithm 4.1** (Parameter canonicalisation). Given a parameter space $\mathcal{W}_h$, proceed:

1: **procedure** CANONICALISE($w = (a_1, b_1, c_1, \ldots, a_h, b_h, c_h, d) \in \mathcal{W}_h$)
2:     ▷ *Stage 1: Reduce the parameter, zeroing out redundant hidden units*     ◁
3:     $Z \leftarrow \{\}$     ▷ *keep track of 'zeroed' units*
4:     **while** any of the following four conditions hold **do**
5:         **if** for some hidden unit $i \notin Z$, $a_i = 0$ **then**     ▷ *reducibility condition (i)*
6:             $b_i, c_i \leftarrow 0$
7:             $Z \leftarrow Z \cup \{i\}$
8:         **else if** for some hidden unit $i \notin Z$, $b_i = 0$ **then**     ▷ —— *(ii)*
9:             $d \leftarrow d + a_i \tanh(c_i)$
10:             $a_i, c_i \leftarrow 0$
11:             $Z \leftarrow Z \cup \{i\}$
12:         **else if** for some hidden units $i, j \notin Z, i \neq j, (b_i, c_i) = (b_j, c_j)$ **then**     ▷ —— *(iii)*
13:             $a_j \leftarrow a_j + a_i$
14:             $a_i, b_i, c_i \leftarrow 0$
15:             $Z \leftarrow Z \cup \{i\}$
16:         **else if** for some hidden units $i, j \notin Z, i \neq j, (b_i, c_i) = (-b_j, -c_j)$ **then**     ▷ —— *(iv)*
17:             $a_j \leftarrow a_j - a_i$
18:             $a_i, b_i, c_i \leftarrow 0$
19:             $Z \leftarrow Z \cup \{i\}$
20:         **end if**
21:     **end while**
22:     ▷ *Stage 2: Negate the nonzero units to have positive incoming weights*     ◁
23:     **for** each hidden unit $i \notin Z$ **do**
24:         $a_i, b_i, c_i \leftarrow \text{sign}(b_i) \cdot (a_i, b_i, c_i)$
25:     **end for**
26:     ▷ *Stage 3: Sort the units by their incoming weights and biases*     ◁
27:     $\pi \leftarrow$ a permutation sorting $i = 1, \ldots, h$ by decreasing $b_i$, breaking ties with decreasing $c_i$
28:     $w \leftarrow (a_{\pi(1)}, b_{\pi(1)}, c_{\pi(1)}, \ldots, a_{\pi(h)}, b_{\pi(h)}, c_{\pi(h)}, d)$
29:     ▷ *Now, $w$ has been mutated into the canonical equivalent parameter*     ◁
30:     **return** $w$
31: **end procedure**

The following theorem establishes the correctness of Algorithm 4.1.

**Theorem 4.2.** *Let $w, w' \in \mathcal{W}_h$. Let $v = $ CANONICALISE$(w)$ and $v' = $ CANONICALISE$(w')$. Then (i) $v$ is functionally equivalent to $w$; and (ii) if $w$ and $w'$ are functionally equivalent, then $v = v'$.*

*Proof.* For (i), observe that $f_w$ is maintained by each iteration of the loops in Stages 1 and 2, and by the permutation in Stage 3. For (ii), observe that Stage 1 isolates functionally equivalent *and irreducible* subparameters $u \in \mathcal{W}_r$ and $u' \in \mathcal{W}_{r'}$ of the input parameters $w$ and $w'$ (excluding the zeroed units). We have $f_u = f_w = f_{w'} = f_{u'}$, so by the results of Sussmann (1992), $r = r' = \text{rank}(w)$, and $u$ and $u'$ are related by unit negation and exchange transformations. This remains true in the presence of the zero units. Stages 2 and 3 are invariant to precisely such transformations. $\square$

# 5 Full Functional Equivalence Class

Algorithm 4.1 produces a consistent output for all parameters within a given functional equivalence class. It serves as the basis for the following characterisation of the full functional equivalence class.

The idea behind the characterisation is to enumerate the various ways for a parameter's units to be reduced, negated, and sorted throughout Algorithm 4.1. Each such *canonicalisation trace* corresponds to a simple set of parameters that takes exactly this path through the algorithm, as follows.

**Definition 5.1** (Canonicalisation trace). Let $r, h \in \mathbb{N}$, $r \leq h$. A *canonicalisation trace of order $r$ on $h$ units* is a tuple $(\sigma, \tau)$, where $\sigma \in \{-1, +1\}^h$ is a sign vector (interpreted as tracking unit negation throughout the algorithm); and $\tau : \{1, \ldots, h\} \to \{0, 1, \ldots, h\}$ is a function with range including $\{1, \ldots, r\}$ (interpreted as tracking unit reduction and permutation throughout the algorithm).

**Theorem 5.2.** *Let $w \in \mathcal{W}_h$ and $v = (\alpha_1, \beta_1, \gamma_1, \ldots, \alpha_h, \beta_h, \gamma_h, \delta) = \text{CANONICALISE}(w)$. Let $r = \text{rank}(w)$. Then the functional equivalence class $\mathfrak{F}[w] \subset \mathcal{W}_h$ is a union of subsets*

$$\mathfrak{F}[w] = \bigcup_{(\sigma,\tau) \in \Gamma(h,r)} \left( X^\delta_{\tau^{-1}[0]} \cap \bigcap_{i=1}^r Y^{\alpha_i, \beta_i, \gamma_i}_{\sigma, \tau^{-1}[i]} \cap \bigcap_{i=r+1}^h Z_{\sigma, \tau^{-1}[i]} \right) \tag{1}$$

*where $\Gamma(h,r)$ denotes the set of all canonicalisation traces of order $r$ on $h$ units and*

$$X^\delta_I = \left\{ (a_1, b_1, c_1, \ldots, a_h, b_h, c_h, d) \in \mathcal{W}_h \,\middle|\, \begin{array}{c} \forall i \in I, b_i = 0 \text{ and} \\ d + \sum_{i \in I} a_i \tanh(c_i) = \delta \end{array} \right\};$$

$$Y^{\alpha,\beta,\gamma}_{\sigma,I} = \left\{ (a_1, b_1, c_1, \ldots, a_h, b_h, c_h, d) \in \mathcal{W}_h \,\middle|\, \begin{array}{c} \forall i \in I, \sigma_i \cdot (b_i, c_i) = (\beta, \gamma) \\ \text{and } \sum_{i \in I} \sigma_i a_i = \alpha \end{array} \right\}; \text{ and}$$

$$Z_{\sigma,I} = \left\{ (a_1, b_1, c_1, \ldots, a_h, b_h, c_h, d) \in \mathcal{W}_h \,\middle|\, \begin{array}{c} \forall i, j \in I, \sigma_i \cdot (b_i, c_i) = \sigma_j \cdot (b_j, c_j) \\ \text{and } \sum_{i \in I} \sigma_i a_i = 0 \end{array} \right\}.$$

*Proof.* Suppose $w' = (a'_1, b'_1, c'_1, \ldots, a'_h, b'_h, c'_h, d') \in \mathcal{W}_h$ is in the union in (1), and therefore in the intersection for some canonicalisation trace $(\sigma, \tau) \in \Gamma(h, r)$. Then $f_{w'} = f_v = f_w$, as follows:

$$f_{w'}(x) = d' + \sum_{i \in \tau^{-1}[0]} a'_i \tanh(b'_i x + c'_i) + \sum_{j=1}^r \sum_{i \in \tau^{-1}[j]} a'_i \tanh(b'_i x + c'_i) + \sum_{j=r+1}^h \sum_{i \in \tau^{-1}[j]} a'_i \tanh(b'_i x + c'_i)$$

$$= \delta + \sum_{j=1}^r \alpha_j \tanh(\beta_j x_i + \gamma_j) \text{ since } w' \in X^\delta_{\tau^{-1}[0]} \cap \bigcap_{j=1}^r Y^{\alpha_j, \beta_j, \gamma_j}_{\sigma, \tau^{-1}[j]} \cap \bigcap_{j=r+1}^h Z_{\sigma, \tau^{-1}[j]}.$$

Now, suppose $w' \in \mathfrak{F}[w]$. Construct a canonicalisation trace $(\sigma, \tau) \in \Gamma(h, r)$ following the execution of Algorithm 4.1 on $w'$. Set $\sigma_i = -1$ where $\text{sign}(b'_i) = -1$, otherwise $+1$. Construct $\tau$ from identity as follows. In each Stage 1 iteration, if the second branch is chosen, remap $\tau(i)$ to 0. If the third or fourth branch is chosen, for $k \in \tau^{-1}[i]$ (including $i$ itself), remap $\tau(k)$ to $j$. Finally, incorporate the Stage 3 permutation $\pi$: simultaneously for $k \notin \tau^{-1}[0]$, remap $\tau(k)$ to $\pi(\tau(k))$.

Note $\text{CANONICALISE}(w') = v$ by Theorem 4.2. Then $w' \in X^\delta_{\tau^{-1}[0]}$ because $\tau^{-1}[0]$ contains exactly those units incorporated into $\delta$. Moreover, for $j = 1, \ldots, r$, $w' \in Y^{\alpha_j, \beta_j, \gamma_j}_{\sigma, \tau^{-1}[j]}$, because $\tau^{-1}[j]$ contains exactly those units incorporated into unit $j$ of $v$, and $\sigma$ their relative signs ($\beta_j > 0$). Likewise, for $j \in r+1, \ldots, h$, $w' \in Z_{\sigma, \tau^{-1}[j]}$ (which is vacuous if $\tau^{-1}[j]$ is empty). $\square$

**Remark 5.3.** If $w \in \mathcal{W}_h$ is irreducible, then $\text{rank}(w) = h$. For $(\sigma, \tau) \in \Gamma(h, h)$, $\tau$ is a permutation (since the range must include $\{1, \ldots, h\}$). The set of traces therefore corresponds to the set of transformations generated by unit negations and exchanges, as in Sussmann (1992).

**Remark 5.4.** When $\text{rank}(w) = h - 1$, there are, modulo sign vectors and permutations, essentially three canonicalisation traces, corresponding to the three ways of adding an additional unit to a $(h-1)$-unit network discussed by Fukumizu and Amari (2000) and Fukumizu et al. (2019): to introduce a new constant unit or one with zero output, or to split an existing unit in two.

**Remark 5.5.** Similarly, in Şimşek et al. (2021, Definitions 3.2 and 3.3), an $(r+j)$-tuple coupled with a permutation plays the role of $\tau$ in characterising the *expansion manifold*, akin to the functional equivalence class but from the dual perspective of adding units to an irreducible parameter. Şimşek et al. (2021) study a setting without a unit negation symmetry, so there is no need for a sign vector.

# 6 Path Connectivity

In this section, we show that the reducible functional equivalence class is piecewise linear path-connected (Theorem 6.1), and, for parameters with rank at most half of the available number of hidden units, has diameter at most 7 linear segments (Theorem 6.3).

**Theorem 6.1.** *Let $w \in \mathcal{W}_h$. If $w$ is reducible, then $\mathfrak{F}[w]$ is piecewise linear path-connected.*

*Proof.* It suffices to show that each reducible parameter $w \in \mathcal{W}_h$ is piecewise linear path-connected in $\mathfrak{F}[w]$ to its canonical representative CANONICALISE($w$). The path construction proceeds by tracing the parameter's mutations in the course of execution of Algorithm 4.1. For each iteration of the loops in Stages 1 and 2, and for each transposition in the permutation in Stage 3, we construct a multi-segment sub-path. To describe these sub-paths, we denote the parameter at the beginning of each sub-path as $w = (a_1, b_1, c_1, \ldots, a_h, b_h, c_h, d)$, noting that this parameter is mutated throughout the algorithm, but is functionally equivalent to the original $w$ at all of these intermediate points.

1. In each iteration of the Stage 1 loop, the construction depends on the chosen branch, as follows. Some examples are illustrated in Figure 1.

   (i) A direct path interpolating $b_i$ and $c_i$ to zero.
   (ii) A two-segment path, interpolating $a_i$ to zero and $d$ to $d + a_i \tanh(c_i)$, then $c_i$ to zero.
   (iii) A two-segment path, interpolating $a_i$ to zero and $a_j$ to $a_j + a_i$, then $b_i$ and $c_i$ to zero.
   (iv) A two-segment path, interpolating $a_i$ to zero and $a_j$ to $a_j - a_i$, then $b_i$ and $c_i$ to zero.

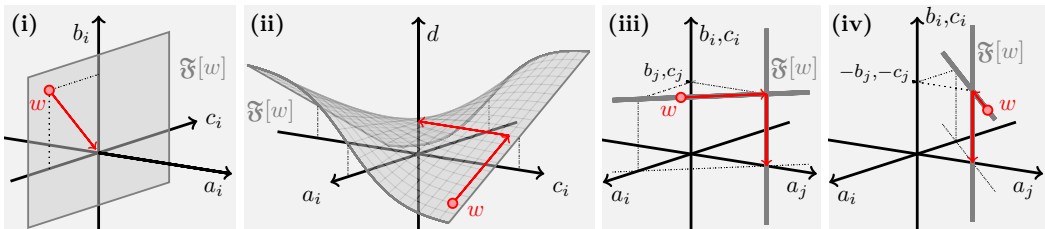

Figure 1: Example paths constructed for each of the Stage 1 branches. Other dimensions held fixed.

Since (the original) $w$ is reducible, (the current) $w$ must have gone through at least one iteration in Stage 1, and must have at least one *blank* unit $k$ with $a_k, b_k, c_k = 0$. From any such parameter $w$, there is a three-segment path in $\mathfrak{F}[w]$ that implements a *blank-exchange manoeuvre* transferring the weights of another unit $i$ to unit $k$, and leaving $a_i, b_i, c_i = 0$: first interpolate $b_k$ to $b_i$ and $c_k$ to $c_i$; then interpolate $a_k$ to $a_i$ and $a_i$ to zero; then interpolate $b_i$ and $c_i$ to zero. Likewise, there is a three-segment path that implements a *negative blank-exchange manoeuvre*, negating the weights as they are interpolated into the blank unit. With these manoeuvres noted, proceed:

2. In each iteration of the Stage 2 loop for which $\text{sign}(b_i) = -1$, let $k$ be a blank unit, and construct a six-segment path. First, blank-exchange unit $i$ into unit $k$. Then, negative blank-exchange unit $k$ into unit $i$. The net effect is to negate unit $i$.

3. In Stage 3, construct a path for each segment in a decomposition of the permutation $\pi$ as a product of transpositions. Consider the transposition $(i, j)$. If $i$ or $j$ is blank, simply blank-exchange them. If neither is blank, let $k$ be a blank unit. Construct a nine-segment path, using three blank-exchange manoeuvres, using $k$ as 'temporary storage' to implement the transposition: first blank-exchange units $i$ and $k$, then blank-exchange units $i$ (now blank) and $j$, then blank-exchange units $j$ (now blank) and $k$ (containing $i$'s original weights).

The resulting parameter is the canonical representative and it can be verified that each segment in each sub-path remains in $\mathfrak{F}[w]$ as required. $\square$

**Remark 6.2.** Şimşek et al. (2021, Theorem B.4) construct similar paths to show the connectivity of their expansion manifold (cf. Remark 5.5). They first connect reduced-form parameters using blank-exchange manoeuvres and then show inductively that each unit addition preserves connectivity.

**Theorem 6.3.** *Let $w \in \mathcal{W}_h$. If* $\mathrm{rank}(w) \leq \frac{h}{2}$, *then* $\mathfrak{F}[w]$ *has diameter at most* 7.

*Proof.* Let $w \in \mathcal{W}_h$ with $\mathrm{rank}(w) = r \leq \frac{h}{2}$. Let $w' \in \mathfrak{F}[w]$. We construct a piecewise linear path from $w$ to $w'$ with 7 segments. By Theorem 6.1, a path exists via the canonical representative parameter $v = \text{CANONICALISE}(w)$. However, this path has excessive length. We compress the length to 7 by exploiting the following opportunities to parallelise segments and 'cut corners'. These optimisation steps are illustrated in Figure 2.

(a) Let the Stage 1 result from Algorithm 4.1 for $w$ be denoted $u$. Let the Stage 1 result for $w'$ be denoted $u'$. Instead of following the unit negation and exchange transformations from $u$ to $v$, and then back to $u'$, we transform $u$ into $u'$ directly, not (necessarily) via $v$.

(b) We connect $w$ to $u$ using two segments, implementing all iterations of Stage 1 in parallel. The first segment shifts the outgoing weights from the blank units to the non-blank units and the output unit bias. The second segment interpolates the blank units' incoming weights and biases to zero. We apply the same optimisation to connect $w'$ and $u'$.

(c) We connect $u$ and $u'$ using two blank-exchange manoeuvres (6 segments), exploiting the majority of blank units as 'temporary storage'. First, we blank-exchange the non-blank units of $u$ into blank units of $u'$, resulting in a parameter $\bar{u}'$ sharing no non-blank units with $u'$. Then, we (negative) blank-exchange those weights into the appropriate non-blank units of $u'$, implementing the unit negation and exchange transformations relating $u$, $\bar{u}'$, and $u'$.

(d) The manoeuvres in (b) and (c) begin and/or end by interpolating incoming weights and biases of blank units from and/or to zero, while the outgoing weights are zero. We combine adjacent beginning/end segments together, interpolating directly from the start to the end, without (necessarily) passing through zero. This results in the required seven-segment path, tracing the sequence of parameters $w, w^1, w^2, \ldots, w^6, w' \in \mathcal{W}_h$.

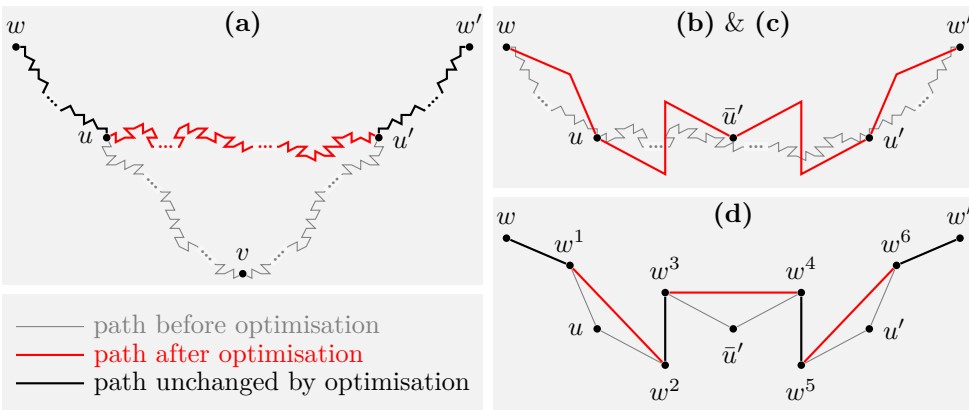

Figure 2: A conceptual illustration of the four path optimisations, producing a seven-segment piecewise linear path of equivalent parameters in a high-dimensional parameter space. **(a)** Follow unit negation and exchange transformations directly between reduced parameters, not via the canonical parameter. **(b)** & **(c)** Parallelise the reduction steps, and use the majority of blank units to parallelise the transformations. **(d)** Combine first/last segments of reduction and blank-exchange manoeuvres.

To describe the constructed path in detail, we introduce the following notation for the components of the key parameters $w, w', u, u', w^1, w^2, \ldots, w^6 \in \mathcal{W}_h$:

$$w = (a_1^w, b_1^w, c_1^w, \ldots, a_h^w, b_h^w, c_h^w, d^w) \qquad u = (a_1^u, b_1^u, c_1^u, \ldots, a_h^u, b_h^u, c_h^u, d^u)$$
$$w' = (a_1^{w'}, b_1^{w'}, c_1^{w'}, \ldots, a_h^{w'}, b_h^{w'}, c_h^{w'}, d^{w'}) \qquad u' = (a_1^{u'}, b_1^{u'}, c_1^{u'}, \ldots, a_h^{u'}, b_h^{u'}, c_h^{u'}, d^{u'})$$
$$w^k = (a_1^k, b_1^k, c_1^k, \ldots, a_h^k, b_h^k, c_h^k, d^k) \qquad (k = 1, \ldots, 6).$$

Of the $h$ units in $u$, exactly $h - r$ are blank—those in the set $Z$ from $\text{CANONICALISE}(w)$. Denote the complement set of $r$ non-blank units $U = \{1, \ldots, h\} \setminus Z$. Likewise, define $Z'$ and $U'$ from $u'$.

With notation clarified, we can now describe the key points $w^1, \ldots, w^6$ in detail, while showing that the entire path is contained within the functional equivalence class $\mathfrak{F}[w]$.

1. The first segment interpolates each outgoing weight from $a_i^w$ to $a_i^u$, and interpolates the output bias from $d^w$ to $d^u$. That is, $w^1 = (a_1^u, b_1^w, c_1^w, \ldots, a_h^u, b_h^w, c_h^w, d^u)$.

   To see that this segment is within $\mathfrak{F}[w]$, observe that since the incoming weights and biases are unchanged between the two parameters, $f_{tw^1 + (1-t)w}(x) = t f_{w^1}(x) + (1-t) f_w(x)$ for $x \in \mathbb{R}$ and $t \in [0,1]$. To show that $f_w = f_{w^1}$, we construct a function $\tau : \{1, \ldots, h\} \to \{0, 1, \ldots, h\}$ from identity following each iteration of Stage 1 of CANONICALISE$(w)$: when the second branch is chosen, remap $\tau(i)$ to 0; and when the third or fourth branch is chosen, for $k \in \tau^{-1}[i]$ (including $i$ itself), remap $\tau(k)$ to $j$. Moreover, we define a sign vector $\sigma \in \{-1, +1\}^h$ where $\sigma_i = -1$ if $\text{sign}(b_i^w) = -1$, otherwise $\sigma_i = +1$. Then:

   $$f_w(x) = d^w + \sum_{j=0}^{k} \sum_{i \in \tau^{-1}[j]} a_i^w \tanh(b_i^w x + c_i^w)$$

   $$= d^w + \sum_{i \in \tau^{-1}[0]} a_i^w \tanh(c_i^w) + \sum_{j=1}^{h} \left( \sum_{i \in \tau^{-1}[j]} \sigma_j \sigma_i a_i^w \right) \tanh(b_j^w x + c_j^w)$$

   $$= d^u + \sum_{j=1}^{h} a_j^u \tanh(b_j^w x + c_j^w) = f_{w^1}(x).$$

2. The second segment completes the reduction and begins the first blank-exchange manoeuvre to store the nonzero units in $Z'$. For $i \in U \cap U'$, pick distinct 'storage' units $j \in Z \cap Z'$. There are enough, as $r \leq \frac{h}{2}$ by assumption thus $|U \cap U'| = |U| - |Z \cap U'| = r - |Z \cap U'| \leq (h - r) - |Z \cap U'| = |Z'| - |Z \cap U'| = |Z' \cap Z|$. Interpolate unit $j$'s incoming weight from $b_j^w$ to $b_i^w$ and interpolate its bias from $c_j^w$ to $c_i^w$. Meanwhile, for all other $j \in Z$, interpolate the incoming weight and bias to zero. This segment is within $\mathfrak{F}[w]$ as for $j \in Z$, $a_j^1 = a_j^u = 0$ by definition of $Z$.

3. The third segment shifts the outgoing weights from the units in $U \cap U'$ to the units in $Z \cap Z'$ prepared in step (2). For $i \in U \cap U'$, pick the same storage unit $j$ as in step (2). Interpolate unit $j$'s outgoing weight from $a_j^u = 0$ to $a_i^u$ and interpolate unit $i$'s outgoing weight from $a_i^u$ to zero. This segment is within $\mathfrak{F}[w]$ as $b_i^2 = b_j^2$ and $c_i^2 = c_j^2$ by step (2).

4. The fourth segment completes the first blank-exchange manoeuvre and begins the second, to form the units of $u'$. For $i \in U'$, interpolate unit $i$'s incoming weight from $b_i^3$ to $b_i^{u'}$ and interpolate its bias from $c_i^3$ to $c_i^{u'}$. This segment is within $\mathfrak{F}[w]$ because for $i \in U' \cap Z$, $a_i^3 = a_i^u = 0$ by definition of $Z$, and for $i \in U' \cap U$, $a_i^3 = 0$ by step (3).

5. The fifth segment shifts the outgoing weights from the selected units in $Z'$ to the units in $U'$ prepared in step (4). We simply interpolate each unit $i$'s outgoing weight to $a_i^{u'}$.

   To see that the segment is within $\mathfrak{F}[w]$, note that $u$ and $u'$ are related by some unit negation and exchange transformations. Therefore, there is a correspondence between their sets of nonzero units, such that corresponding units have the same (or negated) incoming weights and biases. Due to steps (2)–(4) there are $r$ 'storage' units in $w^4$ with the weights of the units of $u$, and the correspondence extends to these storage units. Since the storage units are disjoint with $U'$, this fifth segment has the effect of interpolating the outgoing weight of each of the storage units $j \in Z'$ in $w^4$ from $a_i^u$ to zero (where $i$ is as in step (3)), while interpolating the outgoing weight of its corresponding unit $k \in U'$ from zero to $\pm a_i^u = a_k^{u'}$ (where the sign depends on the unit negation transformations relating $u$ and $u'$).

6. The sixth segment completes the second blank-exchange manoeuvre and begins to reverse the reduction. For $i \in Z'$, interpolate unit $i$'s incoming weight from $b_i^5$ to $b_i^{w'}$, and interpolate its bias from $c_i^5$ to $c_i^{w'}$. This segment is within $\mathfrak{F}[w]$ as for $i \in Z'$, $a_i^5 = a_i^{u'} = 0$ by definition of $Z'$.

7. The seventh segment, of course, interpolates from $w^6$ to $w'$. To see that this segment is within $\mathfrak{F}[w]$, note that by steps (5) and (6), $w^6 = (a_1^{u'}, b_1^{w'}, c_1^{w'}, \ldots, a_h^{u'}, b_h^{w'}, c_h^{w'}, d^{u'})$ (noting $d^u = d^{u'}$ since the output unit's bias is preserved by unit transformations). So the situation is the reverse of step (1), and a similar proof applies. $\qquad\square$

# 7 Discussion

In this paper, we have investigated functional equivalence classes for reducible single-hidden-layer hyperbolic tangent network parameters, and their connectivity properties. Recall that irreducible parameters have discrete functional equivalence classes described by simple unit negation and exchange transformations. In contrast, reducible functional equivalence classes form a complex union of manifolds, displaying the following rich qualitative structure:

- There is a central discrete constellation of *reduced-form* parameters, each with maximally many blank units alongside an irreducible subparameter. These reduced-form parameters are related by unit negation and exchange transformations, like for irreducible parameters.
- Unlike in the irreducible case, these reduced-form parameters are connected by a network of piecewise linear paths. Namely, these are (negative) blank-exchange manoeuvres, and, when there are multiple blank units, simultaneous parallel blank-exchange manoeuvres.
- Various manifolds branch away from this central network, tracing in reverse the various reduction operations (optionally in parallel). Dually, these manifolds trace methods for *adding* units (cf., Fukumizu and Amari, 2000; Fukumizu et al., 2019; Şimşek et al., 2021).

Theorem 6.3 establishes that when there is a *majority* of blank units, the diameter of the entire union of manifolds becomes a small constant number of linear segments. With fewer blank units it will sometimes require more blank-exchange manoeuvres to traverse the central network of reduced-form parameters. Future work could investigate efficient implementation of various permutations through parallel blank-exchange manoeuvres with varying numbers of blank units available.

**Towards modern architectures.** Single-hidden-layer hyperbolic tangent networks appear far from relevant to modern deep learning architectures. However, we expect our canonicalisation algorithm to partially generalise and some aspects of our connectivity findings to generalise as follows.

*Structural redundancy.* Observe that reducibility conditions (i)–(iii) apply to structural redundancies that are generic to every feed-forward layer within any architecture (units with zero, constant, or proportional output). Unit negation symmetries are characteristic of odd nonlinearities only, but other nonlinearities will exhibit their own affine symmetries that would play a similar role. In more sophisticated architectures, these basic sources of structural redundancy will sit alongside other sources such as interactions between layers, attention blocks, layer normalisation, and so on.

*Canonicalisation.* It follows that Algorithm 4.1 serves as a partial canonicalisation for more sophisticated architectures (with small modifications for alternative nonlinearities). Future works need only find and remove the *additional* kinds of structural redundancy.

*Connectivity.* Similarly, additional sources of redundancy expand the functional equivalence class. While global connectivity properties may be lost in this process, individual paths will not be disturbed. We expect that our high-level qualitative finding will hold: that the more reducible a parameter is, the more intricately connected it is to functionally equivalent parameters.

**Towards approximate structural redundancy.** Our framework is built around the definition of functional equivalence, which requires exact equality of functions for all inputs. A more pragmatic definition would concern *approximate* equality of functions for *relevant* inputs. One step in this direction is studying proximity in parameter space to low-rank parameters, though detecting such proximity precisely is formally intractable (Farrugia-Roberts, 2023).

**Functional equivalence and the loss landscape.** Functionally equivalent parameters have equal loss. Continuous directions and piecewise linear paths within reducible functional equivalence classes therefore imply flat directions and equal-loss paths in the loss landscape. More broadly, the set of low- or zero-loss parameters is a union of functional equivalence classes. If some (very) reducible parameters obtain low loss (such as in the overparameterised setting) then the set of low-loss parameters contains rich functional equivalence classes as subsets.

Of course, having the same loss does not require functional equivalence. Indeed, Garipov et al. (2018) observe functional non-equivalence in low-loss paths. The exact relevance of reducible parameters to these topics remains to be clarified. Of special interest is the connection to theoretical work involving unit pruning (Kuditipudi et al., 2019) and permutation symmetries (Brea et al., 2019).

**Canonicalisation and model compression.** Our canonicalisation algorithm transforms a neural network parameter into another parameter that has the same input–output behaviour but effectively uses fewer units. This size reduction is not fundamental to the goals of canonicalisation (we could still achieve canonicalisation by producing a standard, maximally dense representation of each equivalent parameter). Nevertheless, Algorithm 4.1 performs (lossless) model compression as a side-effect (cf. Farrugia-Roberts, 2023).

In particular, our canonicalisation algorithm is reminiscent of a unit-based pruning technique (cf., e.g., Hoefler et al., 2021). However, there are a few salient differences. First, pruning algorithms are usually not designed to preserve exact functional equivalence on all inputs, but rather to accept small changes in outputs (on specific inputs). Second, unit-based pruning techniques usually select individual units for removal, and then continue training the network. Our canonicalisation algorithm instead considers operations that simultaneously remove a unit and perform a systematic adjustment to the remaining weights and biases to maintain the network's behaviour without any training.

Of interest, Casper et al. (2021) empirically studied a network pruning that finds units with weak outputs or pairs of units with correlated outputs, and then *eliminates* or *merges* these units and makes appropriate adjustments to approximately maintain performance. The elimination and merging operations bear a striking resemblance to the operations in Algorithm 4.1.

**Reducible parameters and singular learning theory.** When a reducible parameter is a critical point of the loss landscape, it is necessarily a *degenerate* critical point (due to the continuum of equal-loss equivalent parameters nearby; or cf. Fukumizu, 1996). This places situations where learning encounters (or even approaches) reducible parameters within the domain of singular learning theory (cf. Watanabe, 2009; Wei et al., 2022).

The nature of the degeneracy depends on the exact ways in which it is reducible. For example, is the parameter at the intersection of multiple of the manifolds comprising the functional equivalence class (cf. Theorem 5.2)? The order of variation in directions away from the functional equivalence class also plays a role (cf. Lau et al., 2023). Future work could analyse the structural redundancy to find principled bounds on effective dimensionality.

# 8 Conclusion

Reducible parameters exhibit structural redundancy, in that the same input–output function could be implemented with a smaller network. While reducible parameters comprise a measure zero subset of the parameter space, their functional equivalence classes are much richer than those of irreducible parameters. Understanding these rich functional equivalence classes is important to understanding the nature of the loss landscape within which deep learning takes place.

We have taken the first step towards understanding functional equivalence beyond irreducible parameters by accounting for various kinds of structural redundancy in the setting of single-hidden-layer hyperbolic tangent networks. We offer an algorithmic characterisation of reducible functional equivalence classes and an investigation of their piecewise linear connectivity properties. We find in particular that the more redundancy is present in a parameter, the more intricately connected is its functional equivalence class.

We call for future work to seek out, catalogue, and thoroughly investigate sources of structural redundancy in more sophisticated neural network architectures; and to further investigate the role these parameters play in deep learning.

## Acknowledgements

Contributions (1), (2), and (3) also appear in MFR's minor thesis (Farrugia-Roberts, 2022, §5). MFR was affiliated with the School of Computing and Information Systems at the University of Melbourne while completing this research, but is currently affiliated with the University of Cambridge.

We thank Daniel Murfet for helpful feedback during this research and on this manuscript.

MFR received financial support from the Melbourne School of Engineering Foundation Scholarship and the Long-Term Future Fund while completing this research.

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

# A    Generalising to multi-dimensional inputs and outputs

In this appendix, we consider a slightly more general family of architectures than that introduced in Section 3. Namely, we consider a family of fully-connected, feed-forward neural network architectures with $n \in \mathbb{N}^+$ input units, $m \in \mathbb{N}^+$ biased linear output units, and a single hidden layer of $h \in \mathbb{N}$ biased hidden units with the hyperbolic tangent nonlinearity. With minor modifications, described in the remainder of this appendix, all definitions, algorithms, theorems, and proofs directly generalise from the case $n = m = 1$ to arbitrary $n$ and $m$.

**Multi-dimensional architecture.**    Let $n \in \mathbb{N}^+$, $m \in \mathbb{N}^+$, and $h \in \mathbb{N}$. Define the generalised parameter space $\mathcal{W}_h^{n,m} = \mathbb{R}^{(n+m+1)h+m}$. The weights and biases of the network's units are encoded in the parameter vector in the format $(a_1, b_1, c_1, \ldots, a_h, b_h, c_h, d) = w \in \mathcal{W}_h^{n,m}$ where for each hidden unit $i = 1, \ldots, h$ there is an *outgoing weight vector* $a_i \in \mathbb{R}^m$, an *incoming weight vector* $b_i \in \mathbb{R}^n$, and a *bias* $c_i \in \mathbb{R}$; and $d \in \mathbb{R}^m$ is an *output unit bias vector* containing one bias value for each output unit. This time, $w$ indexes a multi-dimensional mathematical function $f_w : \mathbb{R}^n \to \mathbb{R}^m$ defined as follows:

$$f_w(x) = d + \sum_{i=1}^{h} a_i \tanh(b_i \cdot x + c_i). \tag{2}$$

Note that we use the same tuple notation and ordering $(a_1, b_1, c_1, \ldots, a_h, b_h, c_h, d)$ but now the $a_i$, the $b_i$, and $d$ all denote multi-component vectors. Accordingly, in Equation (2), $b_i$ and $x$ are now multiplied using the inner (dot) product, rather than scalar multiplication, since they are both vectors in $\mathbb{R}^n$. Moreover, $a_i \in \mathbb{R}^m$ as a vector is to be multiplied by the scalar $\tanh(b_i \cdot x + c_i)$. That is, the sum is over vectors of contributions to output units from each hidden unit.

To generalise the results of the main paper to this setting the first change necessary is to replace all mentions of scalar weights with these vectors of weights, and other similar changes such as reading the literal zero as vector zero where appropriate.

**Signing and sorting incoming weight vectors.**    The lexicographic order on $\mathbb{R}^n$, denoted $\preceq$, is a relation such that for $u, v \in \mathbb{R}^n$, $u \preceq v$ if and only if $u = v$ or, in the first index $i = 1, \ldots, n$ where $u$ and $v$ differ, $u_i < v_i$. From this definition we follow the usual conventions in defining $\prec$, $\succ$, and $\succeq$. Finally, define the *lexicographic sign* of $v \in \mathbb{R}^n$, denoted $\mathrm{sign}_{\mathrm{lex}}(v)$, as follows:

$$\mathrm{sign}_{\mathrm{lex}}(v) = \begin{cases} +1 & (v \succ 0), \\ 0 & (v = 0), \\ -1 & (v \prec 0). \end{cases}$$

The parameter canonicalisation algorithm and some of the other theorems and proofs make repeated use of the signs of incoming weight vectors. The lexicographic sign satisfies the requisite properties of the scalar sign function in these uses and so the second change necessary to generalising the results is to replace uses of $\mathrm{sign}(\cdot)$ with uses of $\mathrm{sign}_{\mathrm{lex}}(\cdot)$.

This lexicographic order relation is of course also a total order (see, e.g., Harzheim, 2005, Theorem 4.1.11). Therefore, it allows one to sort a list of vectors. Sorting units by decreasing incoming weights is a key step in Stage 3 of Algorithm 4.1, and so the third change necessary is to use decreasing lexicographic order ($\succeq$) in this stage.

**Generalising Sussmann's equivalence theorem.**    The proofs in the main paper rely on the results of Sussmann (1992) on the equivalence between reducibility and non-minimality, and the fact that irreducible functionally equivalent parameters are related by unit negation and exchange transformations. Sussmann (1992) studied a setting with multiple input units but only a single output unit. Lemmas A.1 and A.2 generalise these results to the multi-output setting.[1] The final necessary change to generalise the results in the main paper is to replace all references to Sussmann's results with references to Lemma A.1 or Lemma A.2.

---

[1]The proofs reduce the multi-output case to the single-output case, so they still rely on the results of Sussmann (1992). A generalisation similar to Lemma A.1 is given by Fukumizu (1996).

The definitions of unit negation and exchange transformations, reducibility, and non-minimality all generalise to arbitrary $n$ and $m$ with the above-mentioned changes. These definitions are repeated here for convenience.

A *unit negation transformation* is a function $\sigma_i : \mathcal{W}_h^{n,m} \to \mathcal{W}_h^{n.m}$ for $i = 1, \ldots, h$, where

$$\sigma_i(a_1, b_1, c_1, \ldots, a_h, b_h, c_h, d) = (a_1, b_1, c_1, \ldots, -a_i, -b_i, -c_i, \ldots, a_h, b_h, c_h, d).$$

A *unit exchange transformation* is a function $\tau_{i,j} : \mathcal{W}_h^{n,m} \to \mathcal{W}_h^{n,m}$ for $i, j = 1, \ldots, h$, where

$$\tau_{i,j}(a_1, b_1, c_1, \ldots, a_h, b_h, c_h, d) = (a_1, b_1, c_1, \ldots, c_{i-1}, a_j, b_j, c_j, a_{i+1},$$
$$\ldots, c_{j-1}, a_i, b_i, c_i, a_{j+1}, \ldots, a_h, b_h, c_h, d).$$

A parameter $w = (a_1, b_1, c_1, \ldots, a_h, b_h, c_h, d) \in \mathcal{W}_h^{n,m}$ is *reducible* if and only if it satisfies any of the following conditions (otherwise, $w$ is *irreducible*):

(i) $a_i = 0$ for some $i$,

(ii) $b_i = 0$ for some $i$,

(iii) $(b_i, c_i) = (b_j, c_j)$ for some $i \neq j$, or

(iv) $(b_i, c_i) = (-b_j, -c_j)$ for some $i \neq j$.

A parameter $w \in \mathcal{W}_h^{n,m}$ is *non-minimal* if and only if $w$ is functionally equivalent to some $w' \in \mathcal{W}_{h'}^{n,m}$ with fewer hidden units $h' < h$.

**Lemma A.1.** *For $w \in \mathcal{W}_h^{n,m}$, $w$ is reducible if and only if $w$ is non-minimal.*

*Proof.* ($\Rightarrow$): A smaller functionally equivalent parameter can be constructed as follows.

(i) If $a_i = 0$ for some $i$, then hidden unit $i$ fails to contribute to the function. Construct a functionally equivalent parameter $w' \in \mathcal{W}_{h-1}^{n,m}$ with hidden unit $i$ omitted:

$$w' = (a_1, b_1, c_1, \ldots, a_{i-1}, b_{i-1}, c_{i-1}, a_{i+1}, b_{i+1}, c_{i+1}, \ldots, a_h, b_h, c_h, d).$$

(ii) If $b_i = 0$ for some $i$, then hidden unit $i$ contributes only a constant to the function. Construct a functionally equivalent parameter $w' \in \mathcal{W}_{h-1}^{n,m}$ with hidden unit $i$ omitted and the output unit bias vector changed to compensate:

$$w' = (a_1, b_1, c_1, \ldots, a_{i-1}, b_{i-1}, c_{i-1}, a_{i+1}, b_{i+1}, c_{i+1}, \ldots, a_h, b_h, c_h, d + a_i \tanh(c_i)).$$

(iii) If $(b_i, c_i) = (b_j, c_j)$ for some $i \neq j$, then hidden units $i$ and $j$ contribute proportionately. They can be combined into a single unit (say, $j$) with the same incoming weights and bias, and a combined outgoing weight vector. Construct a functionally equivalent parameter $w' \in \mathcal{W}_{h-1}^{n,m}$ accordingly:

$$w' = (a_1, b_1, c_1, \ldots, c_{i-1}, a_{i+1}, \ldots, c_{j-1}, a_j + a_i, b_j, c_j, a_{j+1}, \ldots, a_h, b_h, c_h, d).$$

(iv) If $(b_i, c_i) = -(b_j, c_j)$ for some $i \neq j$, then hidden units $i$ and $j$ contribute in negative proportion. Due to the odd property of $\tanh$ they can be combined into a single unit (say, $j$) with incoming weight and bias vectors $(b_j, c_j)$ and a combined outgoing weight vector. Construct a new parameter $w' \in \mathcal{W}_{h-1}^{n,m}$ accordingly:

$$w' = (a_1, b_1, c_1, \ldots, c_{i-1}, a_{i+1}, \ldots, c_{j-1}, a_j - a_i, b_j, c_j, a_{j+1}, \ldots, a_h, b_h, c_h, d).$$

In all cases, the new parameter $w' \in \mathcal{W}_{h-1}^{n,m}$ has $f_{w'} = f_w$, so $w$ is non-minimal.

($\Leftarrow$): We reduce to the single-output case and apply the result of Sussmann (1992) to show that $w$ satisfies at least one of the reducibility conditions.

To reduce to the single-output case, we introduce some notation. From the function $f_w : \mathbb{R}^n \to \mathbb{R}^m$ define a series of component functions $f_w^{(1)}, f_w^{(2)}, \ldots, f_w^{(m)} : \mathbb{R}^n \to \mathbb{R}$ such that for $x \in \mathbb{R}^n$,

$$f_w(x) = \left( f_w^{(1)}(x), f_w^{(2)}(x), \ldots, f_w^{(m)}(x) \right).$$

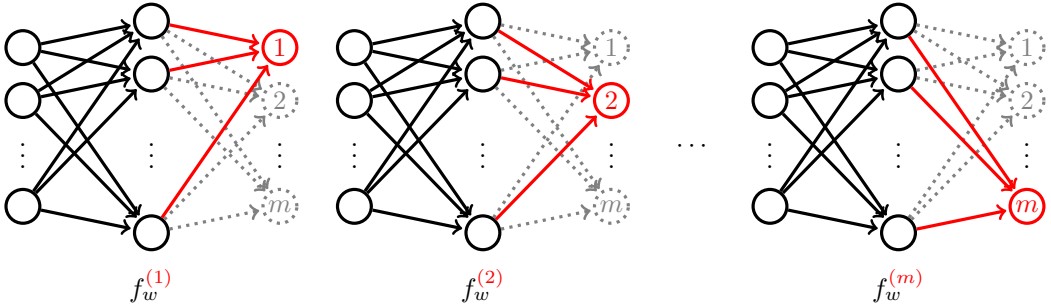

$f_w^{(1)}$ $\qquad\qquad$ $f_w^{(2)}$ $\qquad\qquad$ $f_w^{(m)}$

Figure 3: The connection graphs of the component functions of $f_w$. Included units and weights are solid. The hidden units of each network share the same incoming weights (and biases, not shown).

Each of these component functions is a simple neural network function in an architecture with $n$ input units and $1$ output unit, corresponding to a subgraph of the connection graph of the original neural network, as illustrated in Figure 3.

Denote the corresponding (overlapping) subvectors of $w \in \mathcal{W}_h^{n,m}$ as $w_{(1)}, \ldots, w_{(m)} \in \mathcal{W}_h^{n,1}$. That is, for $\mu = 1, \ldots, m$,

$$w_{(\mu)} = (a_{1,\mu}, b_1, c_1, \ldots, a_{h,\mu}, b_h, c_h, d_\mu) \in \mathcal{W}_h^{n,1}.$$

Now, let $w' = (a_1', b_1', c_1', \ldots, a_h', b_h', c_h', d') \in \mathcal{W}_{h'}$ such that $f_{w'} = f_w$ where $h'$ is the smallest number of hidden units required to implement $f_w$ ($h' < h$ by assumption of non-minimality). Apply the same decomposition to $f_{w'}$ to define $f_{w'}^{(1)}, \ldots, f_{w'}^{(m)}$, and to define $w_{(1)}', \ldots, w_{(m)}' \in \mathcal{W}_{h'}^{n,1}$.

Apply the results of Sussmann (1992) as follows. Since $f_w = f_{w'}$, $f_w^{(\mu)} = f_{w'}^{(\mu)}$ for $\mu = 1, \ldots, m$. It follows that for each $w_{(\mu)}$, $w_{(\mu)}'$ is a functionally equivalent parameter using fewer units. Therefore, the reducibility conditions (in the special case of $m = 1$) must hold for each $w_{(\mu)}$ (Sussmann, 1992).

Since conditions (ii–iv) only depend on incoming weights and biases, if any of these conditions hold for any $w_{(\mu)}$, then they must also hold for $w$ itself (which shares the same incoming weights and biases), and the proof is complete. It remains only to consider the case in which conditions (ii–iv) fail to hold for any $w_{(\mu)}$, and to show that condition (i) holds for $w$ itself in this case.

We must introduce yet further notation. For $i = 1, \ldots, h$ denote by $\varphi_i : \mathbb{R}^n \to \mathbb{R}$ the function $\varphi_i(x) = \tanh(b_i x + c_i)$. Similarly for $j = 1, \ldots, h'$ denote by $\psi_j : \mathbb{R}^n \to \mathbb{R}$ the function $\psi_j(x) = \tanh(b_j' x + c_j')$. Then, since we have ruled out reducibility conditions (ii–iv) for $w$, no $\varphi_i$ is constant (ii) and no two are proportional (iii, iv). The same holds for the $\psi_j$—conditions (i–iv) do not hold for $w_{(\mu)}'$, since $h'$ was assumed to be minimal. Yet, for $\mu = 1, \ldots, m$, the linear combination of functions

$$d_\mu + \sum_{i=1}^{h} a_{i,\mu} \varphi_i - d_\mu' - \sum_{j=1}^{h'} a_{j,\mu}' \psi_j = f_w^{(\mu)} - f_{w'}^{(\mu)} = 0$$

yields the zero function. This linear combination remains when excluding those terms with $a_{i,\mu} = 0$ or $a_{j,\mu}' = 0$. Applying the same reasoning as that in Sussmann (1992), due to the independence property of the hyperbolic tangent function (Sussmann, 1992, Lemma 3.1) the remaining terms must be in bijection, such that

$$\varphi_i = \pm \psi_j \tag{3}$$

for some $j$ with $a_{j,\mu}' \neq 0$ for each $i$ with $a_{i,\mu} \neq 0$.

To complete the proof, note that these relationships (3) between the units of $w$ and $w'$ are independent of $\mu$. However, the relationships are "exclusive" in the sense that no two $\varphi_i$ can be proportional to the same $\psi_j$, else they would also be proportional to each other (ruled out above). Since there are only $h'$ units $\psi_1, \ldots, \psi_{h'}$, it follows that there must be one hidden unit $i$ (actually at least $h - h'$ many units) for which $a_{i,\mu} = 0$ for all $\mu = 1, \ldots, m$ (allowing $\varphi_i$ to avoid any such relationship). That is, $a_i = (a_{i,1}, \ldots, a_{i,m}) = 0$, satisfying condition (i) for $w$ as required. $\qquad\square$

**Lemma A.2.** *Let $w \in \mathcal{W}_h^{n,m}$ be irreducible, and let $w' \in \mathcal{W}_h^{n,m}$. If $w$ and $w'$ are functionally equivalent then there exists a compositional chain of unit negation and exchange transformations, collectively a transformation $T : \mathcal{W}_h^{n,m} \to \mathcal{W}_h^{n,m}$, such that $w' = T(w)$.*

*Proof.* Once again, we reduce to the case $m = 1$ and appeal to Sussmann (1992).

Suppose $w' \in \mathfrak{F}[w]$. Introduce the same decomposition of the two neural networks as in the proof of Lemma A.1, namely, the component functions $f_w^{(1)}, \ldots, f_w^{(m)}, f_{w'}^{(1)}, \ldots, f_{w'}^{(m)}$ implemented by the parameter subvectors $w_{(1)}, \ldots, w_{(m)}, w'_{(1)}, \ldots, w'_{(m)} \in \mathcal{W}_h^{n,1}$ (cf. Figure 3).

For $\mu = 1, \ldots, m$, since $f_w = f_{w'}$, we have that $f_w^{(\mu)} = f_{w'}^{(\mu)}$. Now, $w_{(\mu)}$ and $w'_{(\mu)}$ are not necessarily irreducible, but if they are reducible then it is only by condition (i), since $w_{(\mu)}$ and $w'_{(\mu)}$ have the incoming weights and biases of $w$ and $w'$ respectively ($w$ is irreducible by assumption; $w'$ is irreducible because, with the same number of units as $w$, it is necessarily minimal, and irreducibility follows by Lemma A.1). Remove such units with zero outgoing weight from $w_{(\mu)}$ and $w'_{(\mu)}$ to produce new, functionally equivalent irreducible parameters $u_{(\mu)}, u'_{(\mu)} \in \mathcal{W}_{\mathrm{rank}(w_{(\mu)})}^{n,1}$. Now by Sussmann (1992, Theorem 2.1) there exists a chain of unit negation and exchange transformations $T_\mu$ such that $u_{(\mu)} = T_\mu(u'_{(\mu)})$.

For each $\mu$, $T_\mu$ implies a relationship between the units of $w_{(\mu)}$ and $w'_{(\mu)}$ with nonzero outgoing weights, including possible negations and permutations of these units. This same relationship must hold between those units of $w$ and $w'$ since they share incoming weights and biases with $w_{(\mu)}$ and $w'_{(\mu)}$, and (since $w$ is irreducible, conditions (ii–iv)) these incoming weights are nonzero and the incoming weight and bias vectors are absolutely distinct between units of the same parameter. Moreover, all units are involved in some such relationship because no unit of $w$ or $w'$ can have zero outgoing weight vector by reducibility condition (i).

So, one can construct from these implied relationships a composition of unit negation and exchange transformations relating $w$ and $w'$ as required. □

