# OpenReview forum: "Functional Equivalence and Path Connectivity of Reducible Hyperbolic Tangent Networks"
_NeurIPS.cc/2023/Conference — NeurIPS 2023 poster_

### Official Review · Reviewer_Wcpf · 2023-07-06

**Soundness:** 3 good
**Presentation:** 3 good
**Contribution:** 3 good
**Rating:** 6
**Confidence:** 2

**Summary:**

The paper investigates the functional equivalence class for reducible neural network parameters and its connectivity properties. The authors focus on single-hidden-layer hyperbolic tangent networks, but the findings can be generalized to other feed-forward network components.

**Strengths:**

The paper provides a comprehensive understanding of the functional equivalence class for reducible neural network parameters. This can be key to understanding the structure of the parameter space and the loss landscape on which deep learning takes place.

The authors describe a complex union of manifolds, displaying rich qualitative structure. This includes a central discrete array of reduced-form parameters, connected by a network of piecewise linear paths, and various manifolds branching away from this central network.

The paper establishes that with a majority of blank units, the diameter of this parameter network becomes a small constant number of linear segments. This can be beneficial in understanding the trade-offs between shortest path length and rank for different unit permutations.

The paper also discusses the relevance of their findings to modern architectures and deep learning, suggesting that understanding reducible functional equivalence classes may be key to understanding these topics.


**Weaknesses:**

The paper is highly theoretical and may not provide immediate practical applications for those working with neural networks. The exact relevance of reducible parameters to these topics remains to be clarified.


The paper focuses on single-hidden-layer hyperbolic tangent networks, which may limit the applicability of the findings to more complex architectures.

**Questions:**

In the paper, the authors focus on single-hidden-layer hyperbolic tangent networks. How do the authors think the results would change ifwe considered networks with multiple hidden layers or different activation functions?

The paper discusses the trade-offs between shortest path length and rank for different unit permutations. Could the authors provide more insights into how these trade-offs could impact the performance of neural networks in practical applications?

---

> ### Author Rebuttal · Authors · 2023-08-10
>
> We thank reviewer Wcpf for their thorough review of our submission. We think the strengths section provides a fair summary of the main contributions of the work and that the weaknesses section provides a fair summary of the main scope limitations of the work. We are pleased to reply to the reviewer's questions as follows:
>
> 1. **Beyond single-hidden-layer tanh:** We refer the reviewer to the top-level author rebuttal where we outline a roadmap for generalising our algorithms and analysis beyond the this simple setting. In summary, we think that the results we have presented will form part of the picture in the more general case, but there is additional structure to redundant parameterisations in multiple layers and in other architectures (such as transformer architectures) that is yet to be explored.
>
> 2. **Trade-offs between rank, permutation, and path length and their impacts on learning:** We appreciate this question as the topic seems interesting to us.
>     * We note that the impact on learning of the existence of short paths is through its implications for the structure of the loss landscape. Of interest in particular is the 'local' structure, in this case the number and arrangement of equivalent parameters a small number of path segments away from the current parameter. (See also the discussion in the top-level author rebuttal and the response to some other reviewers).
>     * We have identified two extreme points along this tradeoff:
>         * When the rank of the parameter is one less than the number of units, there are a small number of parameters nearby accessible by a small number of liner segments distance.
>         * When the rank of the parameter is half of the number of units, all equivalent parameters are within a small number of linear segments.
>     * The fundamental driver for these path lengths is the number of 'swap' manoeuvres required to implement the permutation separating the reduced form of two parameters. Some permutations require a large number of serial transpositions to implement, while all permutations can be implemented with a small serial depth if we can perform enough parallel transpositions in each serial step (this parallel transposition approach is at the heart of the diameter result).
>     * The ability to execute transpositions in parallel relies on the number of blank units that can be used as temporary registers for the weigh swap manoeuvres.
>     * Putting it together, this suggests that, the more blank units we have in the reduced form (the lower the rank or the original parameter), the more permutations we will be able to implement with a low equivalent parameters we will be able to implement with a small serial depth (due to more parallelisation), and therefore the more equivalent parameters will be reachable with a small number of linear path segments.

---

> > ### Comment · Reviewer_Wcpf · 2023-08-18
> >
> > Thanks for the author's response.

---

### Official Review · Reviewer_QPQA · 2023-07-06

**Soundness:** 3 good
**Presentation:** 3 good
**Contribution:** 2 fair
**Rating:** 5
**Confidence:** 4

**Summary:**

The paper introduces an algorithm to calculate the canonical equivalent parameter for a tanh feed-forward network.
Further, the full equivalent class of each parameter is given by a union of subsets constructed out of (inverse-)reductions and permutations of the canonical parameter.
Finally, the paper shows that each functional equivalence class is path-connected and that the diameter of a functional class with rank at most the half of the number of units in the hidden layer is 7.


**Strengths:**

The canonicalisation algorithm is novel for feed-forward networks with a tanh transfer function. Furthermore, this is the first result that characterizes the diameter of some of the equivalence classes of these networks.
The paper fully proves each theorem and includes useful visualizations for the path structure of a functional equivalence class.
The paper is overall very clearly written, with only a few possible minor typos.
The results presented here could be of importance to loss minimization problems.

**Weaknesses:**

The scope of applications of the algorithm and the theorems seems to be rather narrow or at least is not properly motivated to be broad.

The paper has very limited evaluation of the proposed algorithm and further uses of the results.

It is for example unclear in which practical application Algorithm 4.1 could be used, because of the real valued parameters being usually all different in a trained network, see Questions.



**Questions:**

196: What is a branch?

265: Should it be ``connect $w'$ to $u$"?

What are the practical applications of this algorithm?
Could you apply it to a loss function example you mention in 354?
If the parameters are real values, wouldn't they (and their absolute values) all be different from each other with probability equal to 1?

Could you say anything about the loss function at different parts of the functional class that belongs to zero loss? Is it steeper at certain points in parameter space than others?

Could you clarify and justify the remark in 362? What is this approximation?

Could you give an example of a network that has rank higher than half of $h$ that is part of a functional equivalence class with a diameter more than 7?

Why does it matter what the diameter is of a functional equivalence class?

Why are networks that are in a functional equivalence class of diameter at most 7 of particular importance?


**Limitations:**

They are adequately discussed.

---

> ### Author Rebuttal · Authors · 2023-08-10
>
> We thank reviewer QPQA for the thorough review of our work. We appreciate in particular the kind comments about the clarity, novelty, and potential importance of the paper.
>
> We hope to be able to sway the reviewer's recommendation to reject the paper by defending the relevance of the results in answer to the reviewer's weaknesses/concerns and specific questions about the motivations. (We are also happy to respond to the reviewer's technical questions.)
>
> **Weaknesses:**
>
> * We refer the reviewer to the top-level author rebuttal on the relevance of the results for a simple architecture. In summary, we believe that the results in this paper are a meaningful component of a generalisation to more practical architectures, and future work can explore the *additional* redundancy structure that emerges as new architectural components are added.
> * We refer the reviewer to our response to **weakness 2** of **reviewer HCLr** for a discussion of the relevance of the canonicalisation algorithm. In summary, this algorithm has clear theoretical utility (for characterising sets of equivalent parameters), which is the main motivation, but also it already has some practical utility for measuring approximate equivalence despite numerical issues and this could be developed further in future work.
> * Regarding the claim in the third question about real weights being unequal probability 1 (which is about the existence/relevance of reducible parameters at all, not just about the canonicalisation algorithm), this depends on the sampling distribution. We view it as unsettled whether weights in trained networks are actually unequal, since they are determined based on data rather than uniformly at random. As a weaker claim, it may also be true that weights are unequal but the course of training is still influenced by the local structure of the parameter space which may involve reducible networks.
>
>     If the reviewer has in mind a direct empirical evaluation of the hypothesis that reducible parameters and their functional equivalence classes are at all relevant to modern deep learning, we agree that such evaluation is an important direction for future work. Motivated by indirect evidence (such as empirical phenomena of compressibility in learned neural networks), we view our contribution as laying the groundwork for a principled study of reducibility and its role in practical deep learning.
>
> **Questions:**
>
> 1. By 'branch' we simply mean 'if' or 'else if' in the algorithm. So the 'second branch' spans lines 8 to 11 of the algorithm (lines 144--147 of the manuscript). We apologise for being unclear. This appears to be standard programming terminology and we are not sure how to clarify it---we welcome specific suggestions.
>
> 2. Yes. Thank you for pointing out this mistake and please accept our apologies for the confusion.
>
> 3. We agree that with floating point parameters the canonicalisation algorithm will not usually face exact equality of any parameters. We refer the reviewer to the second dot point in response to weaknesses above. In summary, the algorithm has theoretical utility and potentially still has practical utility.
>
> 4. It does appear that the steepness of the loss function varies for different equivalent parameters. As a simple example, in a one-unit network *a.tanh(bx)*, if *b* is perturbed around zero for large *a*, the function will vary much faster than if *b* is perturbed around zero for small *a*.
>
> 5. If a parameter *w* achieves loss *L(w)* and is very close in parameter space to some reducible parameter *u*, then *L(u) ~= L(w)* by smoothness of *L*. Then there are paths of parameters with loss equal to *L(u)* throughout the whole functional equivalence class of *u*. In aggregate this implies a path of parameters from *w* to many other parameters with approximately equal loss to *L(w)*. If the reviewer has further questions we will be happy to elaborate.
>
> 6. An example of such a high-diameter set is as follows. Consider a parameter with 10 units but rank 9, meaning there is exactly one 'spare' unit. There is a two-segment (at most) path from this parameter to a 'reduced form' of the parameter where a single unit is blank. Now to reach other equivalent parameters that are separated from this parameter by a permutation, it appears that the permutation must be implemented 'one swap at a time'. Some permutations can be implemented in a single swap. However, other permutations require some large minimum number of serial transpositions to be carried out in order to get all the units into the right place. Implementing one of these permutations to get to an equivalent reduced-form parameter will in general require potentially many more than 7 segments.
>
>     Note: We haven't formally ruled out shorter paths existing than those we construct, which only give upper bounds on shortest path lengths. However, we conjecture that shorter paths do not exist in this architecture based on our experience varying parameters, and we think we could probably prove lower bounds if we invested effort into this. It's not something we have considered yet.
>
> 7. (and 8.) As discussed in the top-level author rebuttal, we do not actually think that the diameter itself or the number 7 are crucially important. Rather, by proving that the diameter is some small constant, we establish that all equivalent parameters are reachable with a small number of path segments. This suggests an extremely tightly connected network of equivalent parameters for these very-reducible parameters. Put another way: what matters is not the low maximum shortest path length but rather than number of short paths.
>
>     In this interest we are attempting to respond to empirical literature observing piecewise linear paths of low loss connecting learned networks (so-called 'mode connectivity' literature). We think our theoretical findings, by identifying one source of such paths, implicate (highly-)reducible networks in these phenomena. See also our Discussion section.

---

### Official Review · Reviewer_HCLr · 2023-07-12

**Soundness:** 3 good
**Presentation:** 3 good
**Contribution:** 3 good
**Rating:** 7
**Confidence:** 3

**Summary:**

This paper focuses on the study of functional equivalence classes in neural networks, specifically in fully-connected, single-layer networks with hyperbolic tangent activation. It  presents a Canonicalisation algorithm for reducible networks to determine canonical representative parameters for different functional equivalence classes. Finally, it explores the concept of connectivity between these classes.

**Strengths:**

The paper gives a well-motivation exploration of functional equivalence classes in neural networks. It provides a well-written definition of these classes, introducing an analysis of connectivity between them. The article references foundational works on simple neural networks and incorporates recent literature in the field of deep learning, enhancing the credibility and relevance of the study.

**Weaknesses:**

The weaknesses of this article stem from the focus on a very theoretical network type with implausible parameters. While a useful theoretical case, further discussion on the applicability of this analysis to modern architectures would benefit the article.

Specifically, a drawback of the article is its narrow focus on a specific architecture type, namely single-layer networks with tanh activation, a single input and single output. While it does discuss expanding the input and output space, it fails to address how the findings extend to more complex multi-layer networks or different activation functions commonly used in modern architectures. By limiting the scope of investigation, the article has limited application to contemporary neural network design and learning.

The proposal of the Canonicalisation algorithm also presents limitations. The algorithm relies on exact equivalence between weights or to 0, which is rarely achievable in practice, especially for weights trained through backpropagation. This raises questions about the adaptability of the Canonicalisation algorithm to networks trained through backpropagation, and whether it can provide meaningful results. The article would greatly benefit from addressing these concerns and exploring potential adaptations for networks trained through backpropagation.

It appears to me that the Canonicalisation algorithm is a unit pruning / neuron removal algorithm. While Kuditipudi et al. is cited, the article would benefit from considering and discussing other neuron removal methods, specifically the following which focus on individual neurons in fully-connected layers:

+ Mao Ye, Chengyue Gong, Lizhen Nie, Denny Zhou, Adam Klivans, and Qiang Liu. Good subnetworks provably exist: Pruning via greedy forward selection. In Proceedings of the 37th International Conference on Machine Learning, pp. 10820–10830. PMLR, 2020.
+ Xiaocong Du, Zheng Li, Yufei Ma, and Yu Cao. Efficient network construction through structural plasticity. IEEE Journal on Emerging and Selected Topics in Circuits and Systems, 9(3):453–464, 2019.
+ Lemeng Wu, Bo Liu, Peter Stone, and Qiang Liu. Firefly Neural Architecture Descent: a General Approach for Growing Neural Networks. In Advances in Neural Information Processing Systems, volume 33, 2020.

Additionally, GradMax is a method which adds new neurons, similarly to Fukumizu and Amari, which was cited. The new neurons do not impact the function of the network upon addition by setting fan-in weights to 0. This case doesn't seem covered by remarks 5.3-5.5.
Evci, Utku, et al. "GradMax: Growing Neural Networks using Gradient Information." International Conference on Learning Representations. 2021.

**Questions:**

The practical contribution of the path connectivity analysis was furthermore unclear for me. What does this analysis mean for learning or architecture design? Is it an argument for sparse training or initialization to reach an architecture with a majority of blank units?

I believe the reference format is incorrect; I believe that NeurIPS uses number-based references instead of name, year.

**Limitations:**

The paper does not adequately address the limitations induced by the narrow focus on a highly theoretical architecture. The sections on "towards modern architectures" and "functional equivalence and deep learning" offer insight into the potential application of the analysis to contemporary deep neural networks, but are not comprehensive.

---

> ### Author Rebuttal · Authors · 2023-08-09
>
> We thank Reviewer HCLr for taking the time to thoroughly review our work. We aim to respond here to the several weaknesses and questions raised. Our question for the reviewer is, if the discussion of limitations and practical considerations in the paper were expanded along these lines, would the reviewer reconsider their recommendation to reject the paper, in line with some of the other reviewers who have recommended varying degrees of acceptance while acknowledging similar limitations? Or, is the reviewer's assessment of our contribution that it is fundamentally insufficient?
>
> **Weakness 1 (limited architecture):** Please see the top-level author rebuttal. In summary, we believe that our results form a meaningful step towards answers for more practical architectures.
>
> **Weakness 2 (exact canonicalisation):** We acknowledge the limitation of the canonicalisation algorithm based on exact equivalence. We do think of the canonicalisation algorithm as primarily a theoretical tool (as a basis for the characterisation of the full functional equivalence class), as the reviewer has noted.
>
> We agree that two neural networks trained to approximate functional equivalence via backpropagation are unlikely to be detected as equivalent by this algorithm for numerical reasons. However, we still believe the canonicalisation algorithm has the potential to be developed in a direction that makes it more applicable as a diagnostic tool for this purpose. We offer the following observations:
>
> * After applying canonicalisation to two such networks, if they really are almost functionally equivalent, then their canonicalisations are likely to be close in parameter space, which can be easily detected (e.g. simply computing L_infinity distance).
> * One 'corner case' where the above claim does not hold is when the networks are near to the boundary of the 'canonical region', the slice of parameter space into which the canonicalisation algorithm maps parameters. Then it is possible that by slightly perturbing a network so as to cross this boundary, and then canonicalising the network, the resulting network will appear far away near another boundary of the region. Such networks will be approximately equivalent but not enough to be detected
> * It is simple to compute the L_infinity distance from a parameter to this boundary, which corresponds to finding the nearest reducible parameter.
> * This leads to a reliable test of approximate functional equivalence for parameters that are closer to each other than they are to the boundary of the canonical region.
>
> If the reviewer finds this sketch compelling, then if accepted, we could add a formal version of these observations as an appendix.
>
> **Weakness 3 (missing discussion of pruning, gradmax):** We thank the reviewer for the detailed reference suggestions, which are highly appreciated. We must admit we are not deeply familiar with the literature on pruning, having come to this problem from the perspective of the literature on functional equivalence.
>
> We acknowledge the apparent similarities between canonicalisation (which involves exact neuron removal) and unit-based network pruning, and also gradmax. One difference we have observed, stemming from the practical motivations of the pruning literature, appears to be that pruning methods usually accept an approximation of functionality (as long as loss is mostly concerned) while we have studied a setting of maintaining exact functional equivalence.
>
> We believe that important future work will bridge these settings.
>
> * In our case that means studying approximate functional equivalence. We have another paper under blind review that makes this connection more precise, discussing a similar algorithm to the canonicalisation algorithm under the framing of 'lossless network compressibility' (we also study approximate relaxations of this problem and their computational complexity).
> * On the other hand, we think our systematic study of functional equivalence can inform pruning literature. Most pruning methods we have seen are based on removing units with little unilateral impact on the function output/the loss. Our study shows that sometimes units can be 'merged', but not unilaterally removed. Is reviewer is aware of any pruning approaches that take advantage of such 'higher-order' opportunities for unit removal? The closest we have seen is: Casper et al., 2019, "Frivolous Units: Wider Networks Are Not Really That Wide", discussing merging units with correlated outputs (cf. our reducibility condition (iii)).
>
> **Question 1 (relevance of path connectivity to learning or architecture design):** The main lessons we hope to draw from this kind of analysis is for learning (rather than architecture design). We have summarised in the Discussion our main takeaways in terms of bulk structure of the parameter space, and therefore the loss landscape, of our analysis. The main insight is in the large number of equivalent parameters that are reachable through short, simple paths from any given reducible parameter (see also: discussion in the top-level author rebuttal about how exactly we expect these insights to generalise). The main relevance of these results draws on the fact that in the overparameterised learning setting, many interpolating solutions are reducible, so this rich structure is inherited by the set of zero-loss parameters.
>
> One could explore training methods that encourage reducibility directly, however, we are more interested in understanding whether or not existing learning methods already cause reducibility to emerge. This is one of our main motivations for theorising these sets of parameters: Now that we have characterised these sets, we can (in future work) conduct experiments aiming to observe their role in learning.
>
> **Question 2 (reference format):** The 2023 formatting instructions say "Citations may be author/year or numeric, as long as you maintain internal consistency." We will follow any updated advice on the matter.

---

> > ### Comment · Reviewer_HCLr · 2023-08-21
> >
> > Thank you for your extensive response. After discussion with other reviewers, there are a few points I'd like to discuss. Apologies that this response comes so close to the end of the discussion period.
> >
> > In the overall response, you note the lack of "direct evidence of reducible parameters being encountered or approached during training in practical settings, which would be the basis for their relevance to modern deep learning theory." In my view, this appears similar to the question about exactness in the canonicalisation algorithm. Are there reducible parameters in deep learning, given the use of backpropagation and stochastic gradient descent? Are there more if the definition is expanded to include approximate equivalence?
> >
> > The expansive pruning literature would seem to indicate that there are: parameter-based removal and merging work when using approximate equivalence. The relation between parameters which can be removed without changing the function of the network and *reducible* parameters as used in this article appears worthy of exploration. One important distinction is that this article considers reducibility based only on the network parameters, and not on any distribution of data. The insight from architecture search, pruning, and other neural structure literature is that the information from the data distribution can often be useful, as a neural network is used as a function of a certain distribution. That being said, data-free initialization and pruning methods exist, based on functional analysis of the network parameters:
> >
> > Namhoon Lee, Thalaiyasingam Ajanthan, and Philip H. S. Torr. 2019. SNIP: Single-shot Network Pruning based on Connection Sensitivity. In International Conference on Learning Representations (ICLR). arXiv:cs.CV/1810.02340
> >
> > Namhoon Lee, Thalaiyasingam Ajanthan, Stephen Gould, and Philip H. S. Torr. 2020. A Signal Propagation Perspective for Pruning Neural Networks at Initialization. In International Conference on Learning Representations (ICLR). arXiv:cs.LG/1906.06307
> >
> > Soufiane Hayou, Jean-Francois Ton, Arnaud Doucet, and Yee Whye Teh. 2021. Robust Pruning at Initialization. In International Conference on Learning Representations (ICLR). arXiv:stat.ML/2002.08797
> >
> > The closest works that I'm familiar with to the canonicalization method are the following, which use data-free analysis of network parameters to merge *similar* parameters:
> >
> > Suraj Srinivas and R. Venkatesh Babu. 2015. Data-free parameter pruning for Deep Neural Networks. In British Machine Vision Conference (BMVC). arXiv:cs.CV/1507.06149
> >
> > Ben Mussay, Daniel Feldman, Samson Zhou, Vladimir Braverman, and Margarita Osadchy. 2020. Data-Independent Structured Pruning of Neural Networks via Coresets. In International Conference on Learning Representations (ICLR). arXiv:cs.LG/2008.08316
> >
> > The coreset article is particularly close to the proposed canonicalisation, as it removes entire neurons and not only individual weights. The analysis used for coreset identification in this work may be useful here: instead of considering a single distribution, the functional equivalence of the network is analyzed over an arbitrary vector.
> >
> > A full review of pruning and sparse architectures is given here:
> >
> > Hoefler, T., Alistarh, D., Ben-Nun, T., Dryden, N., & Peste, A. (2021). Sparsity in deep learning: Pruning and growth for efficient inference and training in neural networks. The Journal of Machine Learning Research, 22(1), 10882-11005.
> >
> > Given this extensive literature and its relevance to the canonical architectures studied in this work, I would appreciate a deeper contextualization of the proposed analysis. The theoretical framework proposed here could be directly linked to these approaches, which have been applied to contemporary deep architectures, thus bringing this work closer to understanding such architectures.
> >
> > Finally, the clarification about the path connectivity analysis is appreciated. In a similar way to my above comments, this approach of studying the learning from canonical architectures appears to me related to the lottery ticket hypothesis:
> >
> > Jonathan Frankle and Michael Carbin. 2019. The Lottery Ticket Hypothesis: Finding Sparse, Trainable Neural Networks. In International Conference on Learning Representations (ICLR). arXiv:cs.LG/1803.03635
> >
> > Do the authors believe that canonical networks are lottery tickets? Could understanding their path connectivity shed light on training sparsely initialized networks?

---

> > > ### Author Response · Authors · 2023-08-21
> > > **Response to Reviewer HCLr**
> > >
> > > **Thank you.** First, thank you so much for these valuable additional literature recommendations. As noted, we are highly interested in exploring the relation between our functional equivalence perspective and the perspective from the literature on pruning and sparsity in deep learning. We appreciate the time you have taken to cite these papers for us.
> > >
> > > **On the relevance of reducible parameters.** We concur about the potential relation between approximate pruning and approximate reducibility, and the value of future exploration of this direction. In the overall response where we mentioned a 'lack of *direct* evidence' for reducible parameters arising or being approached in learning, we indeed had in mind a number of *indirect* sources of evidence for this hypothesis, including:
> > >
> > > 1. the literature on pruning (as you point out) broadly suggesting that learned neural networks are often prunable without dramatic changes in their functionality;
> > > 2. the broader phenomenon of neural network compressibility (beyond pruning), for example the possibility of model distillation through teacher--student learning;
> > > 3. the common use of dropout to enforce redundant functionality within learned networks; and
> > > 4. the lottery ticket phenomenon (see also below).
> > >
> > > **On data-sensitive pruning:**
> > > We appreciate the reviewer pointing out this distinction. What we can say on the matter is that, clearly, in practice we care about a restricted form of functional equivalence that is only sensitive to changes in functionality for realistic inputs. This seems similar to the data-dependent pruning approaches you mentioned. Of course, functional equivalence for all inputs is a sufficient (not necessary) condition for such data-sensitive functional equivalence. This is the case we study, which we agree seems related to data-independent pruning approaches, particularly those based on coresets.
> > >
> > > **On lottery tickets:**
> > > As discussed, the existence of sparse subnetworks containing (most of) a model's functionality appears related to reducibility and canonicalisation. In particular, it seems to us that the existence of a lottery ticket implies that a network is (approximately) reducible. We haven't thought about this particular connection before, but it seems that lottery ticket could potentially be related to canonical parameters, but the connection is not immediate.
> > >
> > > * Roughly, the correspondence would say that the canonical parameter existed as a sparse subnetwork at initialisation. During training, the remaining units would be brought to cancel out so that the behaviour of the sparse subnetwork / canonical parameter could determine the overall function's behaviour. Possibly, the canonical parameter could also distribute its computation by 'splitting' (un-merging) its units into nearby blank units, possibly along paths like those constructed in our paper (e.g. the reverse of the reduction paths).
> > > * To the extent that canonicalisation involves eliminating units that do not contribute, or collections of units that can be merged together and then fail to contribute (because they 'cancel each other'), it seems that the units that are left may look like a sparse subnetwork that performs well before and after training. But another important part of canonicalisation is merging units that each contribute meaningfully to the function. If two such units jointly contribute to the function, it would appear that removing either of them from a subnetwork would meaningfully alter the function. This suggests that finding the hypothetical canonical parameter = lottery ticket after training could be difficult if using pruning methods that do not consider merging proportional units.
> > >
> > > **Overall comments:**
> > >
> > > > The relation between parameters which can be removed without changing the function of the network and reducible parameters as used in this article appears worthy of exploration.
> > >
> > > > Given this extensive literature and its relevance to the canonical architectures studied in this work, I would appreciate a deeper contextualization of the proposed analysis. The theoretical framework proposed here could be directly linked to these approaches, which have been applied to contemporary deep architectures, thus bringing this work closer to understanding such architectures.
> > >
> > > Our above comments are essentially all we have to say so far. If accepted, we would be willing to expand upon the high-level discussion of these related literature(s) in the paper, to better acknowledge this related literature and to call for future work systematically exploring these connections. Unfortunately we are not in a position to promise any detailed exploration of these connections with this submission.
> > >
> > > We thank reviewer HCLr once again for their detailed review, discussion, and literature recommendations, and for their consideration of our submission.
> > >
> > > (Note: unfortunately, we are unavailable for further discussion between now and the end of the discussion period in a few hours.)

---

> > > > ### Comment · Reviewer_HCLr · 2023-08-21
> > > >
> > > > Thank you for the interesting discussion, and apologies again that I replied close to the deadline. I have modified the score of my original review. If the article is expanded to integrate this discussion and to relate the proposed analysis to pruning, sparse networks, and lottery tickets, I think it could be a valuable theoretical basis for further work in deep learning.

---

### Official Review · Reviewer_NhDx · 2023-07-24

**Soundness:** 4 excellent
**Presentation:** 4 excellent
**Contribution:** 4 excellent
**Rating:** 10
**Confidence:** 5

**Summary:**

This paper deals with functional equivalence problems in neural networks, i.e., the characterization
of all neural networks that lead to the same given output function. This is a problem that has been
studied since the early 1990s, including work by the fields medalist Charles Fefferman.
The present paper considers single-layer networks with tanh-nonlinearity and puts forward
a completely new perspective by paying attention to reducible parameters.

**Strengths:**

The paper tackles a problem that has been studied in various guises for over 3 decades, and
puts forward a completely new vantage point, by considering reducible parameters. This leads
to rich insights into the functional equivalence problem. In addition, the paper exhibits a strong algorithmic component,
specifically by providing an algorithmic characterization of redundancies and connecting the underlying
theory to the beautiful concept of piecewise-linear path connected sets.


**Weaknesses:**

could not find any

**Questions:**

it would be interesting to see the authors' thoughts on whether the algorithmic component of the paper can
be extended to multi-layer networks

---

> ### Author Rebuttal · Authors · 2023-08-10
>
> We thank reviewer NhDx for their review of our work and for their generous praise of our contributions. We appreciate the reviewer's acknowledgement of our novel perspective on the functional equivalence problem in particular.
>
> We do acknowledge that our work definitely has important limitations in scope (as discussed in the top-level author rebuttal, and as raised by the other reviewers and acknowledged in our rebuttals). Chief among these limitations is the study of the simple architecture---to the reviewer NhDx's question. We refer the reviewer to our top-level author rebuttal for our detailed thoughts on the path to extending these results to more general architectures.
>
> We will add for the interest of reviewer NhDx that our intuitions in this direction are influenced by the work of Fefferman (studying the *irreducible* multi-layer tanh case) and also the recent work of Vlačić and Bölcskei (e.g., "Affine Symmetries and Neural Network Identifiability", *Advances in Mathematics* 376, DOI 10.1016/j.aim.2020.107485) generalising Sussmann's reducibility conditions substantially (to networks with arbitrary feed-forward connection graphs). We are excited to draw on these works to push beyond the assumption of irreducibility and continue to explore the rich equivalence structure of reducible networks in more general architectures in future work.

---

### Official Review · Reviewer_virJ · 2023-07-25

**Soundness:** 3 good
**Presentation:** 3 good
**Contribution:** 3 good
**Rating:** 5
**Confidence:** 3

**Summary:**

The paper addresses fully connected feed-forward neural networks (NNs) with a single hidden layer and the hyperbolic tangent activation function. A parameter is thus a vector of weights and biases. It considers reducible parameters, which means that a NN with strictly less neurons can implement the same function. The paper provides a "canonicalise" procedure taking a parameter as input and yielding a canonical parameter that implements the same function. Also, if two parameters implement the same function then the output of the procedure is the same for them.

Based on this procedure, the paper characterizes the set of parameters that implement the same function as a given parameter. It shows that this set is piecewise linear path-connected, and if the set is defined with respect to a ``sparse'' parameter, then the number of linear pieces of the diameter of the set is bounded by 7.

**Strengths:**

The paper is well written. The figures are very clear and help understanding.

The theoretical topic is well motivated and well connected to the literature.

The theoretical results are interesting, and so are the proofs in my opinion.

**Weaknesses:**

The main weakness is that the paper is restricted to a single hidden layer.
Also, the hyperbolic tangent activation function is studied, while ReLU would have been more relevant, in my opinion.

For these two weaknesses, it seems they cannot really be fixed in the frame of this paper, as considering multiple hidden layers and/or ReLU would probably change the nature of the results and the proofs.

**Questions:**

The content presented is overall clear.
The statements of the theoretical results seem clear.
Some proofs are a bit technical and more difficult to follow. The authors could consider adding pedagogical content/details on the proofs in the appendix, if they see fit.

**Limitations:**

The authors adequately discuss the main limitation (a single hidden layer) in the discussion.
I do not see potential negative societal impact.

---

> ### Author Rebuttal · Authors · 2023-08-10
>
> We thank reviewer virJ for their thorough review and for their kind words about the clarity of the paper. We appreciate the reviewer's feedback that some of the more technical elements of the proofs were difficult to follow. We are interested in broadening the accessibility of the paper to the extent possible. If possible, we would appreciate if the reviewer noted any specific sections of the proofs that were particularly difficult to follow? This would allow us to efficiently allocate our resources towards clarifying the presentation where it is most needed. Nonetheless, we are exceedingly pleased to hear that the reviewer found our results and proofs interesting.
>
> The main concern of the reviewer appears to be the limited relevance of the setting studied in the paper. We refer the reviewer to the top-level author rebuttal where we have outlined how we see the results as forming an important component of a more general study of redundancy in more modern architectures.
> It's true that the results and proofs will have to change in future work in this direction. However, we contend that these changes will be more like 'expansions' than fundamental changes. New methods will of course be needed, but we believe our contributions will still be useful.
> We will reiterate that with this paper we believe we have identified a self-contained sub-problem (fully analysing the simple architecture) which fits (with clear presentation) into a single conference paper; and that we feel addressing the additional complexity arising from other architectural extensions can be best achieved with separate submissions.
>
> We hope that this defense of the limited scope of the paper might earn the reviewer's reconsideration of their 'borderline' recommendation to accept the paper. However, either way, once again, we thank the reviewer for their attention and consideration.

---

### Author Rebuttal · Authors · 2023-08-09

We thank the reviewers for their time spent thoroughly reviewing our submission. We have responded individually, and we wanted to take the opportunity here in the author rebuttal to expand on the discussion of limitations in the paper.

**Limitations of architecture generality:** All reviewers raised questions or concerns about the limitations in the scope of the study. As the reviewers have acknowledged, we discuss these limitations in the paper, but we are happy to expand here, and if accepted, to expand this discussion in the paper itself.

The main concern appears to be that the results for simple architecture (single-hidden-layer hyperbolic tangent networks) are not relevant to modern practice (with much more complex architectures), and it is unclear how our results will inform a study of the more practical setting.

Concretely, we are interested in generalising the main results (canonicalisation algorithm, characterisation of functional equivalence class, construction of piecewise linear paths) to architectures with features such as ReLU activation, multiple feed-forward layers, residual connections, or even other arrangements of neurons such as attention heads / transformer blocks.

Our claims are as follows:

1. A 'framing'-level contribution of our paper is a readily-generalisable set of questions for studying canonicalisation and functional equivalence beyond irreducible parameters. This appears to be a novel contribution to the theoretical literature.

2. **Our technical results on canonicalisation and functional equivalence are indeed meaningfully informative for future work** studying analogous questions in more complex architectures, as follows:
    * When it comes to functional equivalence in a single hidden layer, while our results have studied the tanh case specifically, most of the structure in the results is not specific to tanh. In particular, reducibility conditions (i) through (iii) are generic to feed-forward layers with any activation function, and only (iv) makes special use of the tanh 'odd' property. Other activation functions, such as ReLU, have their own additional symmetries giving rise to analogous conditions, such as positive linear scaling symmetry for ReLU. However, the methods used in algorithms and proofs for the ReLU case will be similar in how they account for reducibility conditions (i), (ii), and (iii). Therefore parts of our algorithms and proofs will generalise.
    * When it comes to studying the multi-layer case, we have made some headway by generalising our results to the single layer case with multiple inputs and outputs, as a multi-layer feed-forward network can be viewed as a composition of such functions. For this reason, part of canonicalisation in multi-layer architectures will involve canonicalising each layer independently, and then one must also account for interactions between layers. The per-layer canonicalisation will involve a straightforward application of our single-layer results, to which further algorithmic and theoretical analysis could be applied to resolve inter-layer redundancy. In other words, our results will form a significant part of the picture for the multi-layer case.
    * Even transformer blocks are composed of 'pieces' that look like feed-forward network layers, and so our results may partially inform generalisations in this direction.

3. When it comes to considering extensions of our path connectivity results, we expect our results to generalise, but not in the strong sense of 'the same connectivity properties will hold for other architectures'. We should clarify this point (here and in the paper):
    * We do *not* have any specific reason to expect that generalisations of the global connectivity properties or O(1) diameter result in other architectures. When additional sources of redundancy are introduced by extending the architecture, they may expand the functional equivalence class in areas that are not connected by piecewise linear paths.
    * However, while these 'global' connectivity results are the framing we use for technical statements, we believe the interesting implications of these results are *the fact that from a given (highly) reducible parameter, there are many equivalent parameters reachable with (a small number of) linear path components.* This 'local' perspective should not be disrupted by the additional of further equivalent parameters to the set, even if they are globally disconnected.
    * Along these lines we can already conjecture that modern architectures have rich 'local' connectivity properties in the functional equivalence classes of their reducible parameters, simply due to the presence of the same kinds of redundancy between neighbouring units within some layer of the architecture.

**Empirical relevance of reducible parameters:** We are grateful for our reviewers for following our motivation for the study of reducible parameters at all. Our reviewers do not appear to have taken major issue with the main gap in our motivation (for the most part), which is that so far we lack direct evidence of reducible parameters being encountered or approached during training in practical settings, which would be the basis for their relevance to modern deep learning theory. Our personal research agenda involves developing experiments to directly test our hypothesis that these parameters are relevant for learning. We consider this a priority over, say, investing more theoretical effort to systematically study new sources of redundancy arising from richer architectures.

---

Overall, we view hyperbolic tangent networks as a 'toy' architecture in which we have given a complete answer for *one part* of the general topic of redundancy in neural networks, and this is a concrete-enough architecture (historically sufficient) that we might also hope to be able to probe the relevance of reducible networks in practice.

Once again we thank our reviewers for their attention and their consideration.

---

### Decision · Program_Chairs · 2023-09-21

**Decision:**

Accept (poster)

**Comment:**

By accounting for various redundancies in the parameterization of shallow neural networks, tanh activation, all different parameterizations implementing a given function are characterized and are further shown to be piecewise-linear connected.

At the beginning of the review process, opinions were somewhat divided whether the conceptual content, i.e., a framework offering general principles for functional equivalence under *reducible* parameters, outweighs the arguably oversimplified application, i.e., network architectures having single hidden layer and one input. After considerable discussions with the authors clarifying certain concerns, the reviewers have found the former to be a solid contribution to the topic of functional equivalence, mentioning further possible connections to existing work, and eventually unanimously voted for acceptance. The authors are encouraged to incorporate the important feedback given by the knowledgeable reviewers.